**Atmospheric observations made at Oliktok Point, Alaska as part of the Profiling at Oliktok Point to Enhance YOPP Experiments (POPEYE) campaign**

Gijs de Boer[1,2], Darielle Dexheimer[3], Fan Mei[4], John Hubbe[4], Casey Longbottom[3], Peter J. Carroll[4], Monty Apple[3], Lexie Goldberger[4], David Oaks[5], Justin Lapierre[5], Michael Crume[5], Nathan Bernard[5], Matthew D. Shupe[1,2], Amy Solomon[1,2], Janet Intrieri[2], Dale Lawrence[6], Abhiram Doddi[6], Donna J. Holdridge[7], Mark D. Ivey[3], Beat Schmid[4], Michael Hubbell[4]

1) Cooperative Institute for Research in Environmental Sciences, University of Colorado Boulder, Boulder, CO, 80304, USA
2) NOAA Physical Sciences Division, Boulder, CO, 80304, USA
3) Sandia National Laboratories, Albuquerque, NM, USA
4) Pacific Northwest National Laboratory, Richland, WA, USA
5) Fairweather, LLC, Anchorage, AK, USA
6) Department of Aerospace Engineering, University of Colorado Boulder, Boulder, CO, USA
7) Argonne National Laboratory, Lemont, IL, USA

Correspondence to: gijs.deboer@colorado.edu

**Abstract.** Between 1 July and 30 September 2018, small unmanned aircraft systems (sUAS),), tethered balloon systems (TBS), and additional radiosondes were deployed at Oliktok Point, Alaska to measure the atmosphere in support of the second special observing period for the Year of Polar Prediction (YOPP). These measurements, collected as part of the "Profiling at Oliktok Point to Enhance YOPP Experiments" (POPEYE) campaign, targeted quantities related to enhancing our understanding of boundary layer structure, cloud and aerosol properties and surface-atmosphere exchange, and provide extra information for model evaluation and improvement work. Over the three-month campaign, a total of 59 DataHawk2 sUAS flights, 52 TBS flights, and 238 total radiosonde launches were completed as part of POPEYE. The data from these coordinated activities provide a comprehensive three-dimensional data set of the atmospheric state (air temperature, humidity, pressure, and wind), surface skin temperature, aerosol properties, and cloud microphysical information over Oliktok Point. These data sets have been checked for quality and submitted to the US Department of Energy (DOE) Atmospheric Radiation Measurement (ARM) program data archive (http://www.archive.arm.gov/discovery/) and are accessible at no cost by all registered users. The primary dataset DOIs are 10.5439/1418259 (DataHawk2 measurements; Atmospheric Radiation Measurement Program, 2016b), 10.5439/1426242 (TBS measurements; Atmospheric Radiation Measurement Program, 2017) and 10.5439/1021460 (radiosonde measurements; Atmospheric Radiation Measurement Program, 2013a).



## 1. Introduction

Recent decades have seen notable shifts in Arctic climate (Serreze et al., 2007; Screen and Simmonds, 2010). Reductions in sea ice (Maslanik et al., 2011; Comiso et al., 2008), evident as an integrator of a warming Arctic atmosphere (Dobricic et al., 2016; Graversen et al., 2008), and evolving surface energy budget (Mayer et al., 2016; Hudson et al., 2013) act to enhance absorption of solar radiation at the surface due to a dramatic shift in surface albedo (REFS), potentially enhancing Arctic warming. Sea ice reductions also present opportunities for commerce, including natural resource extraction, shipping, and fishing (Smith and Stephenson, 2013; Ho, 2010). Finally, these changes have direct implications on border security due to reduced difficulties with navigation in Arctic waters.

In recognition of the importance of these changes and our need to be able to predict and understand them, several nations have established Arctic atmospheric observatories. These observatories measure atmospheric state, cloud properties, aerosols, winds, and surface meteorology, providing critically needed datasets for assimilation into numerical weather prediction models and to advance the physical understanding of the Arctic atmosphere. In northern Alaska, the US Department of Energy (DOE) Atmospheric Radiation Measurement (ARM) Program currently operates two such observatories. The first is the long-term North Slope of Alaska (NSA) site located in Utqiagvik, which has operated since the late 1990s. Additionally, since 2013, the DOE ARM program has operated its third ARM mobile facility (AMF-3) at Oliktok Point, Alaska. Consortia such as the International Arctic Systems for Observing the Atmosphere (IASOA, Uttal et al., 2016) have formed to support the efficient synthesis of measurements from these and other observatories around the Arctic.

These observatories only represent a fraction of the work to improve our ability to predict the Arctic environment. Groups such as the World Weather Research Programme (WWRP) Polar Prediction Project (PPP) have developed concentrated efforts to support such work. An example of such an effort is the Year Of Polar Prediction (YOPP), taking place from mid-2017 through mid-2019, which directly targets the improvement of prediction capabilities across a wide variety of time scales, from hours to seasons, through coordinated and intensive observations and focused modeling activities. During the "core phase" of the YOPP, two "special observing periods" (SOPs) were conducted in 2018. This includes one SOP in spring (1 February 2018 to 31 March 2018) and one in late summer (1 July 2018 to 30 September 2018). The "core phase" will be followed by a three-year "consolidation phase", during which a variety of experiments and analysis projects will leverage the datasets collected during the core phase to evaluate and improve models, conduct data denial experiments, and evaluate the state of polar prediction.

Based on the input of the global weather and climate modeling communities, YOPP has established a set of detailed modelling priorities, including:

- *Boundary layer including mixed phase clouds*
- *Sea ice modelling*
- *Physics of coupling, including snow on sea ice*



• *High resolution modelling including ensembles*
• *Model validation and intercomparison*
• *Upper ocean processes*
• *The stratosphere*
• *Chemistry, including aerosols and ozone*
As part of the second SOP, the DOE ARM program supported efforts to enhance observational
coverage of the atmosphere at the AMF-3 in Oliktok Point, Alaska (Figure 1).  This project, titled
"Profiling at Oliktok Point to Enhance YOPP Experiments" (POPEYE) included deployment of the
DataHawk2 unmanned aircraft system, tethered balloon systems, and one additional radiosonde
per day (three launches daily) to provide measurements needed to help meet the objectives
above.  The lower-atmospheric thermodynamic observations offer a detailed look into the Arctic
summer time boundary layer providing  insight into its structure  and  evolution, and a means of
validating  retrieval  algorithms  from  remote  sensors.   Such  measurements  support  the  stated
YOPP goal of pursuing an integrated modeling framework to connect cloud, boundary layer and
surface  energy  exchange  schemes  through  Large  Eddy  Simulation  (LES)-based  development.
Additionally,  POPEYE  provides  a  detailed  dataset  that  can  be  used  for  evaluation  of  model
performance  across  a  variety  of  model  products  (e.g.,  reanalyses,  weather  forecast  models,
coupled  regional  forecast  models,  global  climate  models),  and  more  frequent  radiosondes  can
help assess the impact of data assimilation on operational models.  This facilitates studies on the
impact  of  enhanced  Arctic  observations  on  predictions  of  lower  latitude  weather  (e.g.,  Jung,
2014; Inoue et al. 2015). The measurements collected can also provide constraints on the initial
and  boundary  conditions  for  intercomparisons  of  single- column  and  large  eddy  simulation
models.   The  increased  frequency  of  radiosonde  launches  provides  an  enhanced  look  into  the
Arctic  stratosphere,  further  supported  by  the  launch  of  additional  radiosondes  at  other
observatories during this SOP. Finally, POPEYE aerosol measurements provide information on the
vertical structure of key particle properties.
This paper describes the dataset collected during POPEYE.  Section two includes information on
the systems and sensors used, sampling strategies employed, limitations related to weather and
other  factors,  and  a  general  overview  of  the  dataset  as  collected.   Section  three  provides
background  on  the  data  processing  and  quality  control  measures  applied  to  the  datasets
collected during POPEYE, and information on the different levels of data resulting from this effort.
Section four provides information on the availability of the data, including a link for where the
datasets can be downloaded.  Finally, section five provides a summary of the POPEYE campaign.
**2. Description of Measurements and Sampling Strategy**
POPEYE featured a focused deployment of three observational tools during the second northern
hemispheric YOPP SOP.  These measurements were designed to complement measurements
from the instruments integrated into the AMF-3, which run continuously and are therefore not
described in detail in this paper. The reader is referred to comprehensive information available
through the ARM web page (www.arm.gov). The three datasets  described here are those that
were specifically deployed as a part of POPEYE, including the DataHawk2 small unmanned aircraft



system (sUAS), two tethered balloon systems (TBSs) and extra radiosondes. All systems were
deployed by DOE ARM operators, and the Datahawk2 and TBS systems have been deployed
regularly at Oliktok Point over the past few years (de Boer et al., 2018). Here we provide
information on these systems and the sensors operated on each.

### 2.1. Tethered Balloon Systems

TBSs mainly consisted of two different balloons, a 35 m$^3$ helikite constructed by Allsopp Helikites
and a 79 m$^3$ aerostat constructed by SkyDoc$^{TM}$. The helikite is a balloon/kite hybrid that uses
lighter-than-air principles to obtain its initial lift, and a kite to achieve stability and dynamic lift,
while the larger aerostat uses a skirt instead of a kite to achieve stability in flight. Lift of both a
helikite and an aerostat increase with increasing wind speed, so a relatively stable float altitude
can be achieved even in elevated wind speeds. For POPEYE operations, both systems were
operated using an electric winch integrated into a dedicated balloon trailer by Sandia National
Laboratories. The payload and operating guidelines for the TBSs vary significantly with location
and environmental conditions. Generally, the aerostat is operated for total payload weights of 8
– 27 kg, and the helikite is operated for total payload weights < 27 kg. The helikite is not typically
operated above 600 m AGL, because beyond this altitude the weight of the tether and payload
exceed the maximum lifting force of the helikite. The aerostat can be operated at higher
altitudes, but due to its larger size is not launched in sustained surface wind speeds > 7 m s$^{-1}$. The
helikite is not launched in sustained surface wind speeds > 11 m s$^{-1}$. Operation of either platform
is suspended, and the balloon is immediately retrieved if sustained wind speeds at the altitude
of the balloon exceed 15 m s$^{-1}$. In general, the strength of the wind is the main limiting factor
governing the launch and final altitude of the TBSs, with rime accretion on the tether,
instruments and balloon also contributing to altitude limitations.

POPEYE TBS operations involved a variety of sensors and payloads. To measure the
thermodynamic properties of the atmosphere, the TBS team operated multiple different sensor
packages from interMet. This includes the interMet iMet-1-RSB radiosonde package as well as
the interMet XQ2 sensor packages developed for use on UAS. Additionally, a Silixa XT distributed
temperature sensing (DTS) system was flown. This system, which includes a long fiberoptic cable
suspended along the tether, provides a high resolution, continuous measurement of air
temperature based on Raman scattering (Keller et al. 2011; DeJong et al. 2015). Using this
system, the temperature is typically measured along the length of the optical fiber every 30 to
60 seconds at 0.65 cm spatial resolution. To provide information on the winds aloft, vaned cup
anemometers from APRS World were operated at specified intervals along the tether. It is
important to note that while wind speed from these sensors appears to be relatively accurate
when compared with Doppler lidar measurements, a variety of factors including the high latitude
location make the directional measurement inaccurate. Information on the aerosol particle
population was provided using a combination of two Handix Scientific Printed Optical Particle
Spectrometers (POPS) and a TSI Condensation Particle Counter (CPC) 3007. The two POPS
provide information on the aerosol size distribution for particles between 140-3000 nm while the
CPC provides information on the total number of particles between 10-1000 nm. Additionally,
vibrating wire sensors from Anasphere and the University of Reading provide information on the



amount of supercooled liquid water in cloud. These sensors were collectively referred to as
"Supercooled Liquid Water Content" (SLWC) sensors. Further details on all of these sensors and
the expected level of accuracy (where available) are included in Table 1.
The main role of the TBS in POPEYE was to collect detailed information on the vertical structure
of the lower atmosphere over the AMF-3. This provides information on stratification and the
temporal evolution of the lower atmospheric structure. Additionally, the TBS is unique in that it
is able to fly in and above cloud for extended time periods, providing an opportunity to collect
in-situ measurements of thermodynamic, aerosol and cloud microphysical properties on low-
altitude Arctic clouds. To accomplish this, the TBS was flown as high as weather conditions would
permit, conducting repeated profiles with sensors distributed along the tether. While the exact
placement of the sensors would change from flight to flight to adapt to the present conditions,
in general the system was operated with a cluster of sensors including a POPS, CPC, iMet and
SLWC near the top of the tether under the balloon, a DTS fiber along the entire length of the
tether, and subsequent iMet sensors and anemometers below the main package as most
desirable based on the meteorological conditions. When flying the aerostat, a second POPS
would also be flown to get more detailed measurements of evolution of the aerosol profile in
time. A schematic outlining this strategy is included in Figure 2.

### 2.2. DataHawk2 sUAS

Another instrument platform used during POPEYE was the Datahawk2 sUAS, developed at the
University of Colorado Boulder (description of the first version of the DataHawk can be found in
Lawrence and Balsley, 2013). The DataHawk2 sUAS is a small (1.2 m wingspan, <1 kg take- off
weight), robotic, pusher-prop aircraft designed to operate in a variety of conditions as a flexible
and inexpensive measurement platform (see Table 2 for the specifications of the DataHawk2
UAS). The DataHawk2 has been used for a variety of purposes, including the study of turbulence
(e.g. Kantha et al., 2017; Balsley et al., 2018) and high latitude (e.g. de Boer et al., 2016; 2018)
deployments. The relatively slow flight speed (14 m/s, burst up to 22 m/s) allows the platform
to obtain measurements at high spatial resolution when compared to other aerial vehicles.
Despite this relatively slow speed, the DataHawk2 has been operated in winds up to 12 m/s,
making it a robust research platform for the harsh Arctic environment. DataHawk2 flights
completed under POPEYE were generally autopilot guided except for during take-off and landing,
when they were under the control of a local pilot through real-time telemetry. All flights were
completed within radio communication range and within sight of the ground operators and were
conducted within restricted airspace (R-2204, see Figure 1, de Boer et al., 2016) controlled by the
US DOE. This allowed operators to adjust the flight plan in real time to meet the needs of the
science objectives and adapt to the changing environment. The ground controller and UAS
communicate via 2.4 GHz radio with a range of approximately 10 km. Regulations limit
DataHawk2 flight to within visual line of sight, meaning that it is not allowed to fly into clouds
and follow VFR weather minimums for operation (14 CFR 91.155). Additionally, winds hamper
the operation of the DataHawk2, with DOE ARM guidelines restricting flight when winds top 7 m
s$^{-1}$.



The DataHawk2 carries a variety of sensors to make measurements of the atmospheric and
surface states. Custom instrumentation includes a fine wire sensor employing two cold- and one
hot-wire. These provide high-frequency (800 Hz) information on temperature and fine scale
turbulence. High bandwidth is enabled by small surface-area-to-volume ratios of very thin (5 μm
diameter) wires. In addition, the DataHawk2 carries a custom configuration that includes
integrated-circuit slow response sensors (Sensiron SHT) for measurement of temperature
through a calibrated semiconductor, and relative humidity using a capacitive sensor. For
information on surface and sky temperatures, DataHawk2s are equipped with up- and
downward-looking thermopile sensors. These sensors undergo a calibration using targets of a
known temperature. Finally, DataHawk2s have also carried the commercially-available iMet1
radiosonde package, providing comparative information on position (GPS), temperature (bead
thermistor), pressure (piezoresistive) and relative humidity (capacitive).
The main objective for the DataHawk2 was to obtain as many profiles as possible of the lower
atmosphere during daytime hours.  To do this, the aircraft was programed to climb from the
surface to the maximum obtainable altitude.  This maximum altitude was constrained by the
pilot's ability to maintain visual contact with the aircraft (1000 m AGL) or by the cloud ceiling.
Because the endurance of the aircraft is approximately 50 minutes in Arctic operating conditions,
the aircraft could generally complete between one and two full profiles before needing to land
to change batteries.  Because of the substantial interest in the interplay between thermodynamic
and dynamic properties near cloud base, during cloudy conditions, the operators were requested
to hold altitude around cloud base for 10-15 minutes to collect statistics of that environment
before descending back towards the surface.  Figure 3 provides an illustration outlining this flight
pattern.
**2.3. Radiosondes**
The DOE ARM program launched Vaisala RS-92 radiosondes on a regular schedule under POPEYE.
Due to concerns about operator safety and fatigue, the number of radiosondes launched was
scheduled at three per day, with requested launch times of 05:30, 17:30 and 23:30 UTC (21:30,
09:30, 15:30 AKDT) to match the 06:00, 18:00 and 00:00 UTC synoptic times.  Radiosonde
launches were at times suspended due to dangerous conditions, including the presence of bears
on site, or high winds (>13.5 m s$^{-1}$ sustained and gusting >18 m s$^{-1}$) which could result in damage
to the sensor package if the balloon does not achieve enough vertical lift due to the strong cross
wind. Radiosondes are lifted using 350g balloons with an average ascent rate target of 5.5 ms$^{-1}$.).
Radiosonde data from the campaign are available through the ARM data archive (Atmospheric
Radiation Measurement program, 2013a).
**2.4. Overview of meteorological conditions sampled**
The presence of the ARM AMF-3, allows us to put the measurements from the radiosondes, TBS
and UAS in broader context.  Figure 4 shows measurements from the AMF-3 surface
meteorological instrumentation (Atmospheric Radiation Measurement Program, 2013b) over the
three-month POPEYE period.  Synoptically, this period featured several driving features.  For
much of the campaign, there was a stationary area of high pressure positioned over the Gulf of
Alaska, and Oliktok Point sat on the gradient between this area of high pressure and transient



low pressure systems moving through the Chukchi and Beaufort Seas. This generally resulted in
west-northwesterly winds during this time period. Some of these cyclones passed closer to
shore, thereby directly impacting the Oliktok Point area and creating precipitation events and
shifting wind regimes (e.g. July 7-10; August 13; August 16-17; August 29-31). In late August
there was a general shift in the pattern with high pressure beginning to set up over northern
Alaska and eventually over the Beaufort Sea to the north. This resulted in a general shift towards
easterly winds at the surface. The end of the POPEYE campaign featured a dominant area of high
pressure over the area, resulting in weak easterly winds.
Considering the vertical structure of the lower atmosphere, the observations included
measurements from a variety of stability regimes. While the presence of the sun in summer
months generally results in more adiabatic lower atmospheric states than during other times of
year in the Arctic, the data collected indicates sampling of both well-mixed and stratified
conditions. This includes several stable boundary layer cases. Additionally, many of the
completed flights were flown with some level of cloud cover in place. While the UAS did not
sample through the cloud, the TBS was able to do so, providing insight into the thermodynamic
and microphysical structure in and around these clouds. Based on ceilometer data from the AMF-
3 (Atmospheric Radiation Measurement Program 2013c), a cloud base was detected during 76%
of the campaign period. Of the times when clouds were detected, 73% of the cloud bases
occurred below 1 km altitude, 21% occurred between 1-4 km altitude, and 6% were found above
4 km.
In general, it is relevant and important to note that to some extent all of the POPEYE platforms
were weather-limited in terms of their operations. Therefore, there is an element of selective
sampling to consider when using the collected datasets. Most directly, the TBS and UAS systems
were generally not operated during high winds. The UAS additionally had limitations related to
visibility. The radiosondes were least impacted, though high winds did also prevent some
launches.
**2.5. Overview of completed flights and radiosonde launches**
Over the three-month period, there were limited data outages and challenges related to the
issues discussed in the previous sections. Figure 5 illustrates the operations completed under
POPEYE. The most significant challenge to continuous operations was the electromagnetic
interference (EMI) caused by a US Air Force radar station at Oliktok Point. Modifications made
to this radar during the POPEYE time window unfortunately resulted in the grounding of the
DataHawk2s for their planned second and third deployments. Additionally, this EMI resulted in
some resets of the TBS instrumentation, and errors in the TBS GPS readings. In addition, there
were some challenges associated with the Arctic weather. Despite it being summer, winds were
a challenge to both TBS and UAS flights at times, and also resulted in the cancellation of some
radiosonde launches. Wildlife also posed challenges, as the site is visited by both brown and
polar bears during the summer months. The local presence of these large creatures generally
required that operators ceased outdoor operations, impacting all three measurement platforms.
Despite these challenges, the campaign totaled 238 radiosondes launched, 52 TBS flights (134.3



flight hours), and 59 DataHawk2 flights (64.6 flight hours).  Figure 6 illustrates the completed
flights in time-height space.
A map indicating the horizontal extent of the TBS flights is shown in Figure 7 (top).  The horizontal
distances covered are governed by the positioning of the winch trailer for the system, the wind
speed, and the amount of tether extended.  The points drifting over the ocean surface are the
result of erroneous GPS data, likely linked to EMI from the USAF radar system.  The distribution
balloon altitudes (the highest sampling height for any given TBS operation) is shown in Figure 7
(bottom) and demonstrates that the balloon typically sampled the lowest 1 km of the
atmosphere.  Because the balloon can hover at a given altitude for extended time periods, there
are multiple peaks in the altitude distribution, notably at around 150 m, 300 m, 700 m and 1000
m.  These altitudes correspond to altitudes chosen for extended sampling during the campaign.
Also, a comparison of TBS altitudes with ceilometer-based cloud base measurements indicates
that the TBS was operating at or above the lowest detected cloud base altitude 32% of the time.
A map of the horizontal extent of the DataHawk2 flights is shown in Figure 8 (top).  All flights
were conducted in close proximity to the AMF-3 instrumentation, within the restricted airspace
outlined under R-2204.  The flight patterns consisted of profiling of the lowest 1 km of the
atmosphere, as indicated by the probability distribution of altitudes sampled in the lower panel.
This distribution is binned by 20 m increments and based on this it becomes clear that most
common altitude was between 20-40 m above ground level (AGL).  From this altitude, the
frequency of visiting higher altitudes generally decreases slowly, resulting from limitations
imparted by visibility and winds.
Figure 9 provides insight into the statistics of the radiosonde measurements.  The right panel
indicates the distance away from Oliktok Point that radiosondes traveled over the length of the
POPEYE campaign. Within the troposphere (<10 km altitude), radiosondes generally remained
within 20 km of the Oliktok Point facility.  However, a few balloons traveled as far as 100 km away
once in the stratosphere, with most staying within 50 km of the site all the way to the top of the
profile.  The temperature-height histogram (figure 6, left panel) reveals a general cooling of the
air with height through the depth of the troposphere, with most profiles cooling from
temperatures of 0-10 C near the surface to around -50 C at the tropopause.  Additionally, there
are indicators of frequent low-level inversions in the lowest 1-2 km.  There appear to be two
modes of temperatures observed in the stratosphere, with a dominant mode between -40 and -
50 C, and a secondary mode at around -55 C.  Finally, a two-dimensional histogram of the winds
with height (Figure 6, middle panel) illustrates a broad range of measurements near the surface
(0-20 m s$^{-1}$), with winds generally increasing with height through the troposphere to values
ranging between 5-50 m s$^{-1}$.  Winds in the stratosphere again decrease to less than 10 m s$^{-1}$.
Figure 10 illustrates time-height cross sections of radiosonde measurements of temperature,
relative humidity and wind speed for the duration of POPEYE.
**3.  Data processing and quality control**
The US DOE ARM program handles all data collection, quality control, and processing for field
campaigns.  In general, several different levels of ARM data are made available, ranging from raw





data as recorded by the sensors (a-level), to quality-controlled data (b-level) and data products
(c-level).  This section provides an overview of the processing and quality control applied to the
data streams coming from the platforms deployed during POPEYE.
For the DataHawk, current processing techniques provide both raw and processed datafiles.
Aircraft performance and sensor data are gathered and stored in a binary format on the onboard
SD card. This binary format data is the raw data that is archived by ARM (a0 level). Typically, this
raw data is invisible to the community user, but can be requested through the ARM data
discovery tool if desired. In addition, the data on the SD card is unpacked, downsampled to 10
Hz, and assigned to a relevant array of variable names, and then exported to NetCDF format as a
processed raw data file (a1 level). This data file includes data gathered by onboard sensors during
flight, aircraft performance data, telemetry data and GPS data. The next file that is produced is a
10 Hz quality-controlled file that includes some initial conversions (b1 level).  For example, raw
sensor data from the cold wire sensor and onboard temperature sensors are used to convert the
voltage reported by the cold wire into a temperature value.  Additionally, relative humidity and
infrared temperature values measured are calibrated and converted from the engineering to
relevant physical units. Wind components are reconstructed using corrected pitot airspeed data,
GPS data, and the aircraft principle axis data to produce wind speed and direction and the three
wind components. Finally, a quality control step is applied to remove any significant spikes in the
dataset.  This quality-controlled dataset is the current final ARM data product for DataHawk2. An
additional higher frequency data product is under development for future release, which will
provide the turbulence parameters as a value added product (VAP).
Most of the TBS measurements undergo a similar processing and quality control procedure.  In
particular, several quality control measures are implemented on the POPS instrument.  Included
in this processing is a size correction that is determined through routine size checks and
calibration.  For the size check, 500 nm polystyrene latex (PSL) particles are generated to evaluate
the signal response from the POPS instrument and confirm that the instrument performance is
steady over the course of the campaign.  For the calibration, eight different PSL particle sizes are
used to determine the relationship between the optical response signal and particle size.  In
addition, a flow correction is applied, which is based on routine checks using a flow meter.
Radiosonde data are processed as quality-controlled measurements, with quality control being
completed proprietary Vaisala software that corrects for sensor response time and solar
radiation exposure.
**4.  Data Availability**
The data files from POPEYE observations are available for public download through the US DOE
ARM Program Data Archive (http://www.archive.arm.gov/discovery/).  ARM uses NetCDF as the
standard data file format, with  self-describing metadata provided to the user inside the NetCDF
file.  The data are posted as individual datastreams on the archive, which is searchable by site (in
this case OLI for Oliktok Point) and instrument (in this case "TBS" for the tethered balloons,
"aafdatahawk" for the DataHawk2, and "sonde" for the radiosondes).  Each instrument may have
several different levels of data available.



The main TBS datastream for measurements from the iMet instruments and basic information
on aerosol instrumentation is *olitbsimetM1.a1* (DOI: 10.5439/1246367).   ARM is currently
working to produce a quality-controlled b1 product.   Data from the DTS system has been
collected by the ARM Data Management Facility (DMF), and can be requested by email to
armarchive@ornl.gov, with the appropriate DTS datastreams for POPEYE being *tbsdtssxforjch1*,
*tbsdtssxforjch2*, *tbsdtssxch1*, *tbsdtssxch2*. SLW sensor data is available through the ARM archive
under the *tbsslwc.b0* datastream, while the TBS aerosol instrumentation can also be downloaded
through the archive as *tbscpcM1.00*, *tbspopdryM1.00*, *tbspopwetM1.00*.  All of these datasets
are currently provided at 1 Hz.   TBS ground station data, including temperature, humidity,
pressure and winds at the surface, are available as b-level files on the archive under the file prefix
"olitbsgroundM1" as 10-minute average values.
Quality-controlled DataHawk data can be downloaded as *oliaafdatahawkmetU1.b1* (DOI:
10.5439/1426242).  Finally, the POPEYE radiosonde dataset is available as a QC'd b1 dataset, with
the filenames being of the general form olisondewnpnM1.b1 (DOI: 10.5439/1021460), where
wnpn" refers to the mode of the sonde data collection.  Here, "w"=winds, "p"=PTU (pressure,
temperature, humidity), and "n"=nominal indicates a normal flight with data collection during
ascent only.
To make it possible for scientists to cite DOE ARM program data in their publications, ARM
recognizes the value of Digital Object Identifiers (DOIs).  Such DOIs are generally being generated
at the ARM data product level.  Data products produced from the a-level data may have their
own DOI -- for example, separate DOIs are assigned to each of the available output datastreams
and any value-added products (VAP) from the radiosonde measurements obtained by ARM.  This
means that it is possible that POPEYE measurements could be spread across a variety of DOIs,
and that additional DOIs could be created that include POPEYE data as additional data products
are developed.
**5.  Summary**
Between 1 July and 30 September 2018, the POPEYE measurement team collected detailed
measurements of the lower Arctic atmosphere at Oliktok Point, Alaska using tethered balloons,
unmanned aircraft and radiosondes.  This activity resulted in the completion of 134.3 TBS flight
hours, 64.6 sUAS flight hours, and 238 radiosonde launches.  The primary focus of POPEYE was
to provide detailed measurements of the lower atmosphere, including thermodynamic state,
aerosol properties, cloud microphysical properties, winds, and surface temperature.  UAS flights
covered the atmosphere between the surface and 1 km altitude but were unfortunately called-
off early due to EMI from the nearby long-range surveillance radar system operated by the US
Air Force. Tethered balloon measurements went as high as 1396 m using two different balloons.
Radiosondes were launched at a frequency of three times daily, except when environmental
conditions (winds, bears) prevented balloon launches.  These datasets provide a detailed look
into processes in the lower atmosphere and set the stage for detailed evaluation of numerical
models and, together with ongoing, continuous measurements from the AMF-3, support the



development of modeling case studies for process understanding and evaluation of
parameterization performance.
Quality-controlled versions of the data collected as a part of POPEYE are available on the US DOE
ARM data archive. This archive is publicly accessible and allows users to download data from
these platforms and all other ARM-operated instrumentation, including measurements from the
AMF-3 deployment at Oliktok Point.
**Author Contributions**
GdB designed the field campaign, acted as principal investigator for POPEYE, conducted field
work as part of POPEYE, and led the development of the manuscript. DD, CL and MA were the
primary TBS operators during POPEYE, contributed to the processing of TBS data, and contributed
to the writing and review of the manuscript. JH, PC, and LG were the primary DataHawk2
operators during POPEYE and contributed to the processing of DataHawk2 data and the writing
and review of the manuscript. DO, JL, MC and NB are site operators at Oliktok Point and
conducted the radiosonde launches, contributed to site operations during POPEYE and assisted
the DataHawk2 and TBS teams while in the field. FM is the instrument mentor for TBS aerosol
instrumentation as well as for the DataHawk2 and contributed to data preparation and
processing for POPEYE as well as manuscript writing and review. MS, AS, and JI are POPEYE Co-
PIs and contributed to campaign planning, field work, and oversight as well as the writing and
review of this manuscript. DL is the primary DataHawk2 developer and contributed to the
development and review of the DataHawk2 dataset. AD helped with the derivation of wind
estimates from the DataHawk2. DH is the ARM instrument mentor for the radiosondes and
contributed to the processing of the radiosonde dataset as well as the writing and review of this
manuscript. Finally, MI and BS manage the teams responsible for operation of the TBS and
DataHawk2. Additionally, MI is the primary site manager at the AMF-3. They both oversaw and
supported campaign activities and additionally contributed to the review of this manuscript.
**Acknowledgments**
This work was supported by the US Department of Energy Atmospheric Radiation Measurement
Program. Support for campaign planning and execution was provided by the US DOE
Atmospheric System Research Program under project DE-SC0013306. Finally, additional support
was provided by the NOAA Physical Sciences Division. We would like to thank the US Air Force
for providing access to the Oliktok Point facility, ENI Petroleum who supported our teams at their
Nikaitchuq Operations Center, and ConocoPhillips who housed team members at the Kuparuk
camp. Finally, POPEYE is an officially-endorsed contribution to the Year of Polar Prediction
(YOPP), a flagship activity of the Polar Prediction Project (PPP), initiated by the World Weather
Research Programme (WWRP) of the World Meteorological Organisation (WMO). We
acknowledge the WMO WWRP for its role in coordinating this international research activity.






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




**Tables**

|  | Resolution | Accuracy | Range | Response Time |
|---|---|---|---|---|
| **iMet-1-RSB** |  |  |  |  |
| Pressure (hPa) | < 0.01 | +/- 0.5 | 2 - 1070 | < 1 s |
| T (°C) | < 0.01 | +/- 0.2 | -95 to 50 | 2 s |
| RH (%) | < 0.1 | +/- 5 | 0 - 100 | 2 s @ 25 °C |
| GPS Altitude (m) |  | +/- 15 | 0 – 30+ km |  |
| GPS Wind Speed (m/s) |  | +/- 1 |  |  |
| GPS Position (m) |  | +/- 10 |  |  |
|  |  |  |  |  |
| **iMet XQ2** |  |  |  |  |
| Pressure (hPa) | 0.01 | +/- 1.5 | 10 - 1200 | 10 ms |
| T (°C) | 0.01 | +/- 0.3 | -90 to 50 | 1 s @ 5 m/s flow |
| RH (%) | 0.1 | +/- 5 | 0 - 100 | 5.2 s @ 5 °C |
| GPS (m) |  | +/- 12 vertically |  |  |
|  |  |  |  |  |
| **APRS World Wind Vane** |  |  |  |  |
| Wind Speed (m/s) | 0.1 | +/- 0.1 or 5% (whichever is greater) | 1 - 59 |  |
| Wind Direction* (°) | 1 | +/- 2 | 0 - 360 |  |
|  |  |  |  |  |
| **POPS** |  |  |  |  |
| Particles Conc. (cm$^{-3}$) |  | +/- 10 % < 1000 cm$^{-3}$ at 0.1 LPM | 0-1250 cm$^{-3}$ |  |
|  |  |  |  |  |
| **CPC** |  |  |  |  |
| Particles Conc. (cm$^{-3}$) |  | +/- 2.5-3% | 0-1E$^{4}$ cm$^{-3}$ |  |
|  |  |  |  |  |
| **TBS Ground Station** |  |  |  |  |
| T (°C) | 0.01 | +/- 0.3 | -95 to 50 | < 1 s |
| RH (%) | 0.1 | +/- 2 @ 20 °C, < 90% RH, +/- 3 @ 20 °C, >= 90% RH | 0.8 - 100 | 15 s @ 20 °C |

**Table 1:** Known performance characteristics for TBS instruments. The asterisk with wind
direction denotes that these stated specifications have not been met in the Arctic environment
at Oliktok Point.



**Table 2:** Known performance characteristics for DataHawk2 instruments. The asterisk with wind
direction denotes that these stated specifications have not been met in the Arctic environment
at Oliktok Point.

| Data Type | Resolution | Accuracy | Range | Response Time |
|---|---|---|---|---|
| GPS latitude [degrees] | 0.010 | 10m | -40 to 80 | 1s |
| GPS longitude [degrees] | 0.010 | 10m | -180 to 180 | 1s |
| GPS altitude [m, MSL] | 0.010 | 10m | -100 to 15000 | 1s |
| Baro pressure [mbar] | 0.01 | 2.5 | 500 to 1030 | 0.022 s |
| Rel. humidity [%] | 0.01 | +/- 3 | 0 to 105 | 8 s |
| Slow temp. [°C] | 0.015 | +/- 2 | -40 to 80 | 2 s |
| Coldwire Voltage [V] | 0.0000078 [~0.025°C] | Unknown | -40 to +80 °C | 0.5 ms @ 15 m/s |
| Airspeed [m/s] | 0.01 | 0.2 | 0 to 30 | 0.3 ms |
| iMet, EE03, Temp [°C] | 0.01 | +- 0.3 deg | -40 to + 85 deg C | 1s |
| iMet, EE03, RH [%] | 0.01 | +- 3% | 0-95% | 1s |
| wind_u [m/s] | 0.01 | Unknown | -50 to 50 | 0.1s |
| wind_v [m/s] | **0.01** | Unknown | -50 to 50 | 0.1s |
| wind_w [m/s] | **0.01** | Unknown | -50 to 50 | 0.1s |






**Figures**

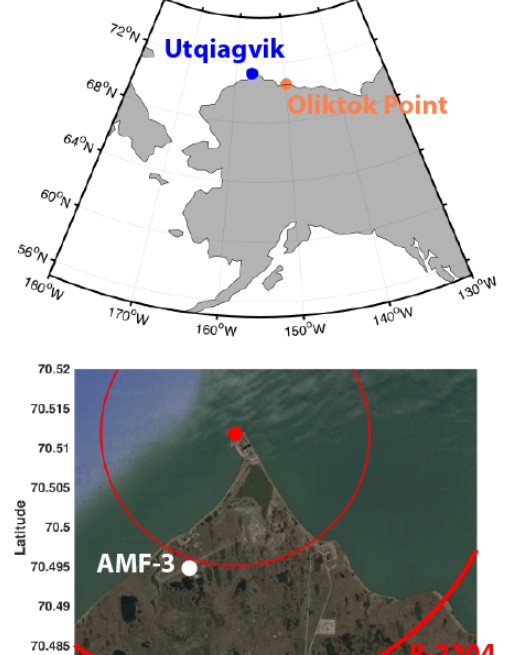

**Figure 1:** A map illustrating the location of Oliktok Point, Alaska (top). The lower panel is a
satellite image of the Oliktok Point area, including information on the boundaries of the R-2204
restricted airspace (bold red line), and the location of the DOE AMF-3 (white dot).



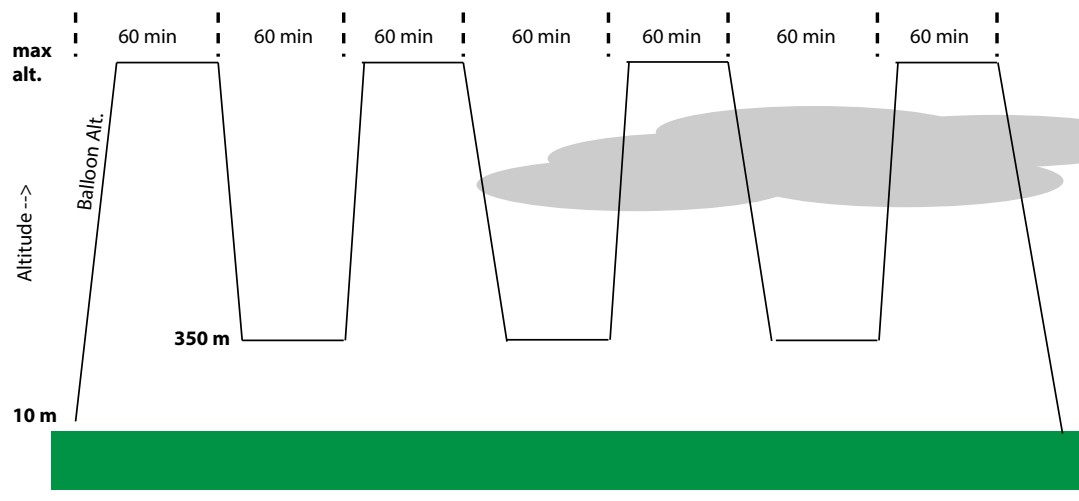

**Figure 2:** An illustration of the proposed TBS flight pattern for clear or cloudy conditions. The
black lines are the proposed flight pattern, with time on the horizontal axis.

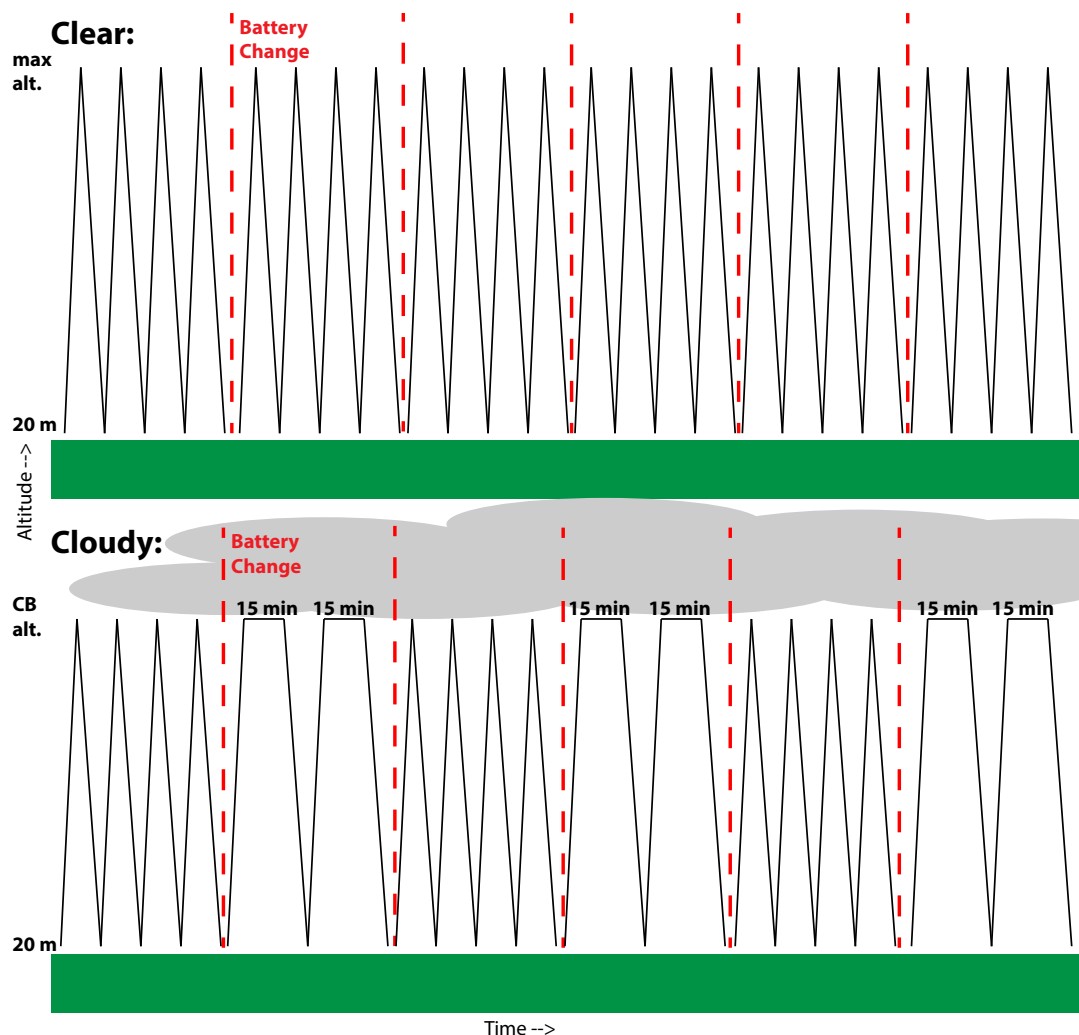

**Figure 3:** An illustration of the proposed DataHawk2 flight pattern for clear (top) and cloudy
(bottom) conditions. The black lines are the proposed flight pattern on a time axis, while the red
lines indicate battery changes in between flights.





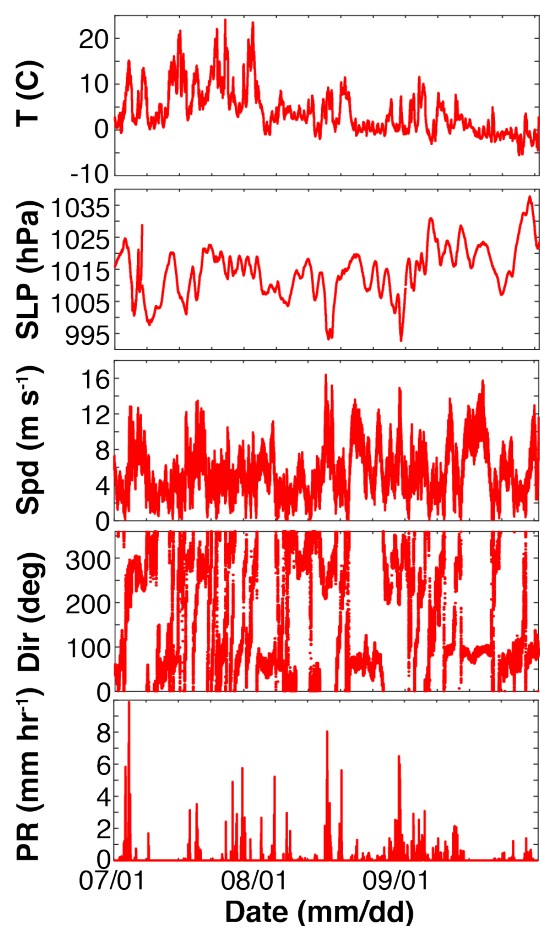

**Figure 4:** Surface meteorological conditions, as measured by instrumentation associated with
the Oliktok Point AMF3 during POPEYE. From top to bottom are: 2-meter air temperature, sea
level pressure, 10-meter wind speed, 10-meter wind direction and surface precipitation rate.



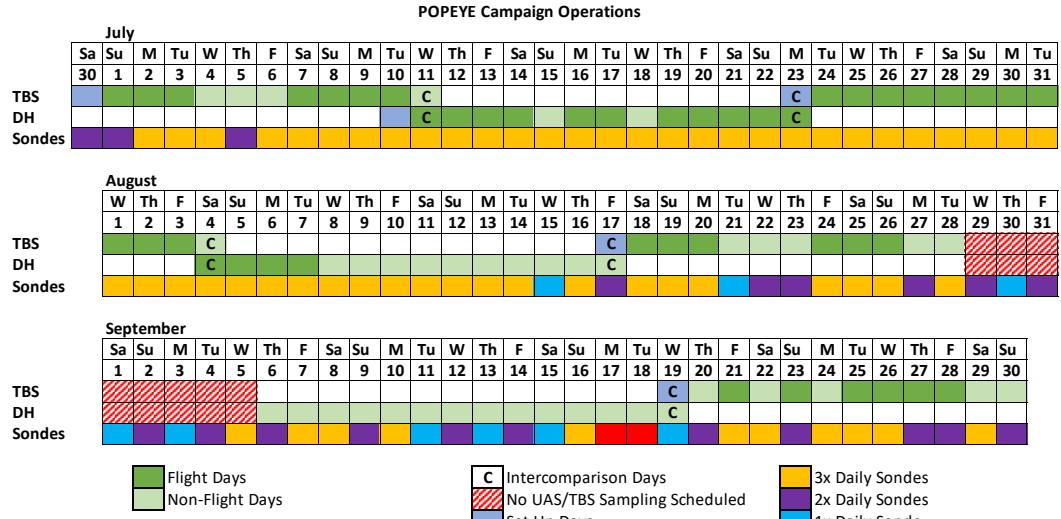

**Figure 5:** A graphical representation of actual UAS, TBS and radiosonde operations during
POPEYE.

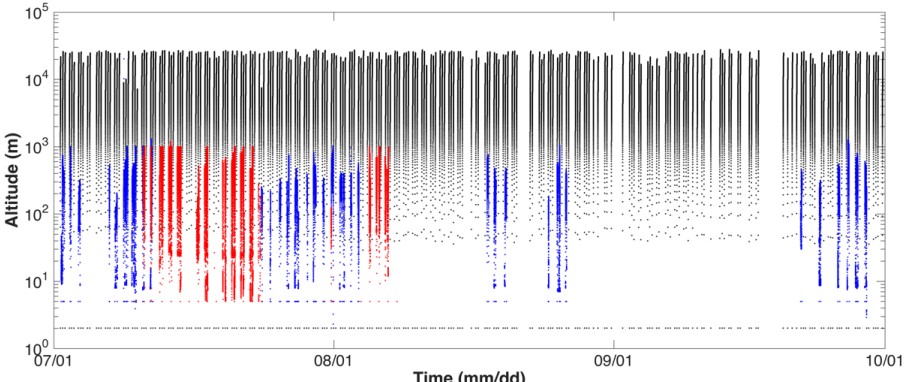


**Figure 6:** A time-height cross-section illustrating all of the POPEYE radiosonde launches (black
dots), DataHawk2 flights (red dots) and tethered balloon flights (blue dots).


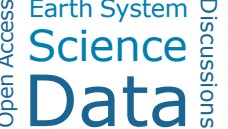



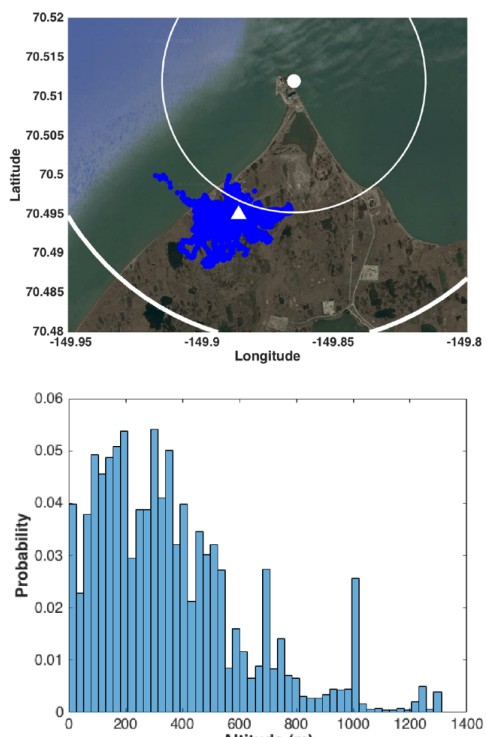

**Figure 7:** POPEYE tethered balloon flight locations (top), including white range rings at one and
two nautical miles demonstrating the extent of R-2204 and the location of the AMF-3 (white
triangle). The bottom panel is a relative frequency distribution of the altitudes sampled by the
TBS during POPEYE.





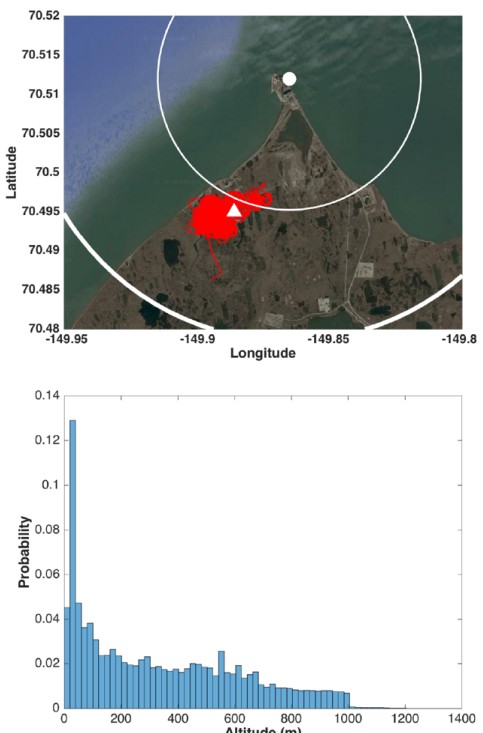

**Figure 8:** POPEYE DataHawk2 flight locations (top), including white range rings at one and two
nautical miles demonstrating the extent of R-2204 and the location of the AMF-3 (white triangle).
The bottom panel is a relative frequency distribution of the altitudes sampled by the DataHawks
during POPEYE.


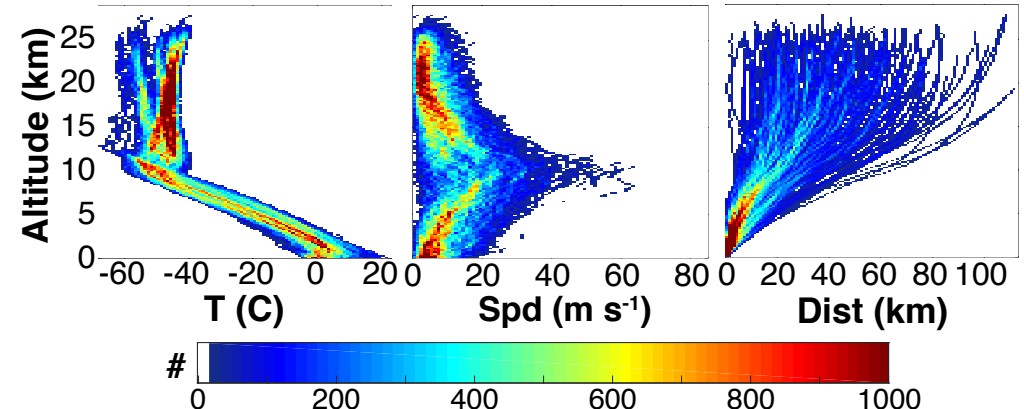

**Figure 9:** Two-dimensional histograms of radiosonde temperature (left), wind speed (middle),
and distance from Oliktok Point (right), with altitude during POPEYE.

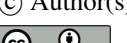

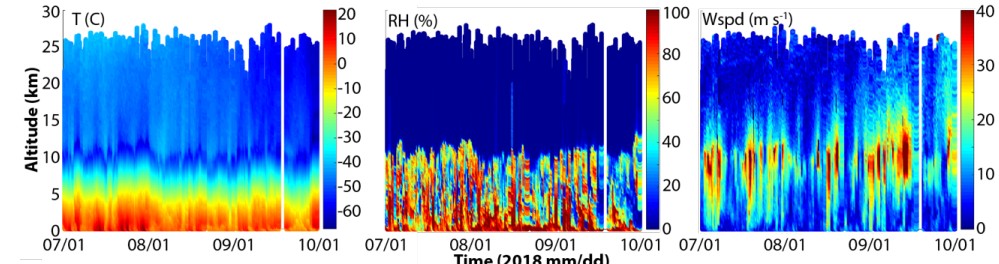

**Figure 10:** POPEYE radiosonde data, including time-height cross sections of (left to right) temperature, relative humidity and wind speed as observed during the second YOPP Special Observing Period.