# Peer review of "Atmospheric observations made at Oliktok Point, Alaska as part of the Profiling at Oliktok Point to Enhance YOPP Experiments (POPEYE) campaign"

_Earth System Science Data, 2019_

## Referee Comment (RC1) · Anonymous Referee #1 · 8 May 2019

I have to admit I was a little perplexed at being asked to review this article that really contain no scientific ideas other than the discussion of the experiment and data, but I see from the journal description that is the purpose of this specific journal.

The paper describes a unique field campaign and the resulting data obtained. Best I can determine the description of the experiment follows the original motivations, and the document adequately describes the processing performed on the raw data. The data described can be found at the sites identified in the paper, and thus is freely available, although it is not clear to me whether the email request for the TBS data

meets the freely available requirement for this journal (I assume it does). Only comment I have is that the authors bring all the labelling in the figures to a standardized format. As is, there is a large degree of variability in font sizes, with some figures nice on the eyes and others with font sizes well below what can be read in print form.

---

## Referee Comment (RC2) · Anonymous Referee #2 · 22 May 2019

The authors describe measurements and provide sound and unique dataset of the lower Arctic atmosphere obtained at Oliktok Point, Alaska using tethered balloons, unmanned aircraft and radiosondes. Personally, I am not a big fan of do-it-yourself science on the base of provided data sets, but I understand somebody might be. The manuscript reads well and fulfils the requirements of the journal. Also data sets are accessible via ARM Data Management Facility as guided in manuscript and obey common standards. I have not found any flaws in presentation quality of the manuscript. I have only minor comments to current version of the manuscript, please see below.

[Figure]

1. Introduction, p3, l89, the following sentence is bit confusing, ". . . and one additional radiosonde per day (three launches daily). . .". Does it mean at least one per day and/or up to three per day?

2.2 DataHawk2 sUAS, p6, l225, How authors determined the cloud base altitude and how close they operated their sUAS to cloud base, could they be more specific?

2.5. Overview of completed flights and radiosonde launches, p7, l280. This is very interesting reading, could authors be bit specific about the distance of take-off area to radar station? Maybe also radar frequency and its power? I understand if authors do not want to share those details, I am just curious.

3. Data processing and quality control, p9, l359. Authors describe quality control of POPS instrument; however nothing is mentioned about TSI 3007 total aerosol sensor. Also by "flow correction", p9, l365, do authors mean flow correction for the height or routine flow calibration at ground level?
* * *

---

## Referee Comment (RC3) · Anonymous Referee #3 · 23 May 2019

The article provides a good overview of the measurement activities at a very important measurement site. More details on the sensors and data quality would be nice, as specified below. In particular for the growing community using unmanned aerial systems, more specific descriptions would be helpful. I suggest to perform minor revisions, as suggested in the following, before publishing the manuscript.

- title: "Atmospheric observations" sounds very general. In the overview, mainly meteorological parameters are presented. What about aerosol? There are aerosol sensors, and aerosol measurements have been done. Why is this not included in the

overview, and at least some profiles are shown? Are the aerosol data included in the data bases? I would suggest modifying the title to know what is meant by "atmospheric observations".

- l. 194: which autopilot was used?

- Section 2.2: please specify exact type of each sensor, plus manufacturer and country of origin. This makes it easier for other users to identify the sensor, and do comparisons.

- l. 223: What is the typical turnaround time between two flights?

- l. 237: remove the bracket at the end

- l. 292/ Fig. 6: Please comment on the obvious gaps in the time series of the measurements

- l. 412: please show some results of the aerosol and cloud microphysical properties as well

- l. 417: up to which wind speed were radiosonde launches possible?

- l. 440/441: "derivation of wind estimates", Table 2 – please provide error bars for wind speed!

- Table 1 and 2: please use the same style, e.g. with units of accuracy, put the caption either on top or at the bottom of the table, provide at least a conservative estimation on wind speed accuracy

- Fig. 4: which temporal resolution of the data is shown? Averaged over 30 min? 1 day?

- Fig. 5: The colour scheme is misleading. I would expect that colous indicate measurement days. What does the colour white mean here? I would mark the flight days, but not the non-flight days. I would further suggest leaving the setup days and the "No

UAS/TBS Sampling Scheduled" white, as there were no measurements. The reader should have an overview of data availability, not on other activities. Please explain why you mention in particular the intercomparison days with a C. What does it mean for the data? Is the data better on this day? Was a new calibration performed?

- Fig. 7/8: Please explain the white dot – probably the launch site? Add in the caption that the balloon flight locations are marked in blue, and the DataHawk flights in red.

- Fig. 9: Very nice and important plot! It would be good to have some more discussion on it. Could you do something similar for aerosol, of course for lower altitudes only?

---

## Author Comment (AC1) · 20 Jun 2019

We thank all of the reviewers for their valuable comments. Below, we have added some responses (in red) to the comments submitted (in black).

**Reviewer 1 Comments:**

I have to admit I was a little perplexed at being asked to review this article that really contain no scientific ideas other than the discussion of the experiment and data, but I see from the journal description that is the purpose of this specific journal.

We thank the reviewer for their comments. While ESSD does not field in depth scientific evaluations, it does serve an important purpose in helping to document the data used by the scientific community and help to recognize the work that goes into the collection of such datasets.

The paper describes a unique field campaign and the resulting data obtained. Best I can determine the description of the experiment follows the original motivations, and the document adequately describes the processing performed on the raw data. The data described can be found at the sites identified in the paper, and thus is freely available, although it is not clear to me whether the email request for the TBS data meets the freely available requirement for this journal (I assume it does). Only comment I have is that the authors bring all the labelling in the figures to a standardized format. As is, there is a large degree of variability in font sizes, with some figures nice on the eyes and others with font sizes well below what can be read in print form.

This is a very good suggestion. We have updated our figures to have common font sizes (10 pt), assuming that figures are either 6.5" wide (full page width) or 3" wide (single column width).

**Reviewer 2 Comments:**

The authors describe measurements and provide sound and unique dataset of the lower Arctic atmosphere obtained at Oliktok Point, Alaska using tethered balloons, unmanned aircraft and radiosondes. Personally, I am not a big fan of do-it-yourself science on the base of provided data sets, but I understand somebody might be. The manuscript reads well and fulfils the requirements of the journal. Also data sets are accessible via ARM Data Management Facility as guided in manuscript and obey common standards. I have not found any flaws in presentation quality of the manuscript. I have only minor comments to current version of the manuscript, please see below.

We would like to thank the reviewer for their time in reading through the manuscript. We have to admit that we don't quite understand the "do-it-yourself science" statement, and it is unclear as to whether it is in response to the material in this paper, or to the potential for someone to use this paper and conduct their own science with the measurements. In either case, as the PIs and executing team for this field campaign, we hope that the measurements are used widely and welcome anyone to access the datasets.

1. Introduction, p3, l89, the following sentence is bit confusing, ". . . and one additional radiosonde per day (three launches daily). . .". Does it mean at least one per day and/or up to three per day?

This statement meant that one additional radiosonde was launched daily during the campaign (on top of the two that are launched at all times. This results in three launches daily. We have updated the text to read "(three launches daily versus the standard twice-daily launch schedule followed at the observatory)

2.2 DataHawk2 sUAS, p6, l225, How authors determined the cloud base altitude and how close they operated their sUAS to cloud base, could they be more specific?

We have updated the text to include "as determined from the observatory ceilometer and visual tracking of the aircraft" and "While the cloud base height is variable, ideally the altitude held by the aircraft would be within 25 m of the mean cloudbase level."

2.5. Overview of completed flights and radiosonde launches, p7, l280. This is very interesting reading, could authors be bit specific about the distance of take-off area to radar station? Maybe also radar frequency and its power? I understand if authors do not want to share those details, I am just curious.

We have added: "located approximately 150-300 m from the primary DataHawk2 flight areas". Regarding the frequency and power, we have not included this in the paper because our knowledge is limited to what can be found on Wikipedia. However, we assume this to be a combination of AN/FPS-117 and AN/FPS-124 radar systems from Lockheed Martin. These are D/L-band (1215-1400 MHz) systems that typically operate at 24.6 kW (117) and an unknown power (124).

3. Data processing and quality control, p9, l359. Authors describe quality control of POPS instrument; however nothing is mentioned about TSI 3007 total aerosol sensor. Also by "flow correction", p9, l365, do authors mean flow correction for the height or routine flow calibration at ground level?

There is no correction applied to account for altitude changes. In terms of general quality control, the TSI 3007 is routinely calibrated to ensure that the flow rate is correct, and ground base comparisons are conducted with other butanol CPC to ensure that these are within 15% of one another in terms of particle concentration. Additionally, daily zero count checks are completed, and the alcohol wick is recharged and replaced as needed. Data is flagged as "questionable", when particle concentrations are higher than $10^5$ particle/cc, because of a lack of correction for coincident sampling at high concentrations. We have added this material into the body of the text.

**Reviewer 3 Comments:**

The article provides a good overview of the measurement activities at a very important measurement site. More details on the sensors and data quality would be nice, as specified below.

In particular for the growing community using unmanned aerial systems, more specific descriptions would be helpful. I suggest to perform minor revisions, as suggested in the following, before publishing the manuscript.

- title: "Atmospheric observations" sounds very general. In the overview, mainly meteorological parameters are presented. What about aerosol? There are aerosol sensors, and aerosol measurements have been done. Why is this not included in the overview, and at least some profiles are shown? Are the aerosol data included in the data bases? I would suggest modifying the title to know what is meant by "atmospheric observations".

We realize that this is a general name but given the fact that we were observing atmospheric thermodynamic state, dynamic state, and aerosol properties, it seemed appropriate to us to have this be wide-ranging. The reviewer makes a good point about not having included enough information on the aerosol measurements. We have added a paragraph in the "overview of completed flights" section on aerosols and added an aerosol-centric figure.

- l. 194: which autopilot was used?

The DataHawk2 uses a custom autopilot developed at the University of Colorado.

- Section 2.2: please specify exact type of each sensor, plus manufacturer and country of origin. This makes it easier for other users to identify the sensor, and do comparisons.

We do not believe that including the country of origin and many details on the sensors will help this paper. While we agree that it is useful to understand what sensor was used and have added some additional details. However, ultimately so much depends on **how** the sensor is integrated, that just having the sensor model number does not necessarily provide useful information to the reader. Therefore, for many readers having exhaustive detail on the sensors would be distracting rather than help to provide insight on the field campaign and the types of measurements that are available.

- l. 223: What is the typical turnaround time between two flights?

Back-to-back flights can be completed with a turnaround time of as little as 10 minutes. Some time is required to change the batteries and re-calibration of the navigation system. We added the following line to the text: "The turnaround time between flights can be as short as 10 minutes, but is generally on the order of 15-30 minutes."

- l. 237: remove the bracket at the end

Thank you for pointing this out – we have removed the extra parenthesis and period.

- l. 292/ Fig. 6: Please comment on the obvious gaps in the time series of the measurements

As mentioned, the tethered balloon and DataHawk2 were scheduled to operate on alternating two-week time periods over the three-month campaign. As discussed in the text, the DataHawk2

(represented by the red points on the map) was grounded half way into its second deployment due to EMI issues related to the long-range surveillance radar at the site.  Therefore, the red dots stop after the middle of the campaign.  The DataHawk2 was scheduled to complete a third two-week deployment in early September, which is why there is a large gap there. Gaps in the tethered balloon periods and radiosonde launches were weather-related.

- l. 412: please show some results of the aerosol and cloud microphysical properties as well

As mentioned above, we have included a figure with aerosol data.  The cloud liquid water content data is not tested enough (or processed enough) to include in a figure at the current time.  We mention it because there may be people interested in analysis of such a measurement.

- l. 417: up to which wind speed were radiosonde launches possible?

The ARM program will launch radiosondes in winds up to 30 mph (13.1 m/s).  We have added this to the text.

- l. 440/441: "derivation of wind estimates", Table 2 – please provide error bars for wind speed!

We have updated this text and have added wind accuracy estimates based on a recent comparison with surface-based instrumentation (Barbieri et al., 2019).

- Table 1 and 2: please use the same style, e.g. with units of accuracy, put the caption either on top or at the bottom of the table, provide at least a conservative estimation on wind speed accuracy

We have updated the tables to have similar styles.  Additionally, we have added rough estimates of wind uncertainties based on recent intercomparison work.

- Fig. 4: which temporal resolution of the data is shown? Averaged over 30 min? 1 day?

The temporal resolution shown is one minute.  We have added this in the caption.

- Fig. 5: The colour scheme is misleading. I would expect that colours indicate measurement days. What does the colour white mean here? I would mark the flight days, but not the non-flight days. I would further suggest leaving the setup days and the "No UAS/TBS Sampling Scheduled" white, as there were no measurements. The reader should have an overview of data availability, not on other activities. Please explain why you mention in particular the intercomparison days with a C. What does it mean for the data? Is the data better on this day? Was a new calibration performed?

We appreciate this feedback. However, we were trying to distinguish between days when the TBS/DataHawk2 were scheduled to fly but couldn't because of weather (light green) and days when they were not scheduled to fly (white).  We agree that the "No UAS/TBS sampling scheduled" dates could be made white as well and have done so in the revised figure.  The intercomparison dates were days on which the teams overlapped, and therefore, some priority was given towards surface-based intercomparison of sensors, as well as spatial and temporal colocation of flights. This resulted in data that can be compared to evaluate whether there are any notable biases between different sensors.

- Fig. 7/8: Please explain the white dot – probably the launch site? Add in the caption that the balloon flight locations are marked in blue, and the DataHawk flights in red.

The white dot is the center of the airspace used. The launch locations varied, but were generally near the AMF-3 (white triangle). We have updated the caption as recommended.

- Fig. 9: Very nice and important plot! It would be good to have some more discussion on it. Could you do something similar for aerosol, of course for lower altitudes only?

We completely agree that the radiosonde data should be explored and discussed in far greater detail, though we believe that such a description would be outside of the scope of an ESSD article (and likely deserves its own article in a journal geared towards scientific analysis of data). We have added a figure for aerosol data from the TBS flights.